# Towards Example-Based NMT with Multi-Levenshtein Transformers

Maxime Bouthors$^{\heartsuit,\clubsuit}$, Josep Crego$^{\clubsuit}$, and François Yvon$^{\heartsuit}$

$^{\heartsuit}$Sorbonne Université, CNRS, ISIR , F-75005 Paris, France
`firstname.lastname@isir.upmc.fr`
$^{\clubsuit}$Systran , 5 rue Feydeau, F-75002 Paris, France
`firstname.lastname@systrangroup.fr`

## Abstract

Retrieval-Augmented Machine Translation (RAMT) is attracting growing attention. This is because RAMT not only improves translation metrics, but is also assumed to implement some form of domain adaptation. In this contribution, we study another salient trait of RAMT, its ability to make translation decisions more transparent by allowing users to go back to examples that contributed to these decisions. For this, we propose a novel architecture aiming to increase this transparency. This model adapts a retrieval-augmented version of the Levenshtein Transformer and makes it amenable to simultaneously edit multiple fuzzy matches found in memory. We discuss how to perform training and inference in this model, based on multiway alignment algorithms and imitation learning. Our experiments show that editing several examples positively impacts translation scores, notably increasing the number of target spans that are copied from existing instances.

## 1 Introduction

Neural Machine Translation (NMT) has become increasingly efficient and effective thanks to the development of ever larger encoder-decoder architectures relying on Transformer models (Vaswani et al., 2017). Furthermore, these architectures can readily integrate instances retrieved from a Translation Memory (TM) (Bulte and Tezcan, 2019; Xu et al., 2020; Hoang et al., 2022), thereby improving the overall consistency of new translations compared to past ones. In such context, the autoregressive and generative nature of the decoder can make the process (a) computationally inefficient when the new translation has very close matches in the TM; (b) practically ineffective, as there is no guarantee that the output translation, regenerated from scratch, will resemble that of similar texts.

An alternative that is attracting growing attention is to rely on computational models tailored to edit existing examples and adapt them to new source

|  |  | precision | % units |
|---|---|---|---|
| unigram | *copy* | 87.5 | 64.9 |
|  | *gen* | 52.6 | 35.1 |
| bigram | *copy-copy* | 81.4 | 55.0 |
|  | *copy-gen* | 40.1 | 8.9 |
|  | *gen-copy* | 39.5 | 10.7 |
|  | *gen-gen* | 34.2 | 25.4 |

Table 1: Modified precision of copy vs. generated unigrams and bigrams for `TM-LevT`. For bigrams, we consider four cases: bigrams made of two copy tokens, two generated tokens, and one token of each type.

sentences, such as the Levenshtein Transformer (`LevT`) model of Gu et al. (2019). This model can effectively handle *fuzzy matches* retrieved from memories, performing minimal edits wherever necessary. As decoding in this model occurs is non-autoregressive, it is likely to be computationally more efficient. More important for this work, the reuse of large portions of existing translation examples is expected to yield translations that (a) are more correct; (b) can be transparently traced back to the original instance(s), enabling the user to inspect the edit operations that were performed. To evaluate claim (a) we translate our test data (details in Section 5.1) using a basic implementation of a retrieval-augmented `LevT` with TM (`TM-LevT`). We separately compute the modified unigram and bigram precisions (Papineni et al., 2002) for tokens that are copied from the fuzzy match and tokens that are generated by the model.[1] We observe that copies account for the largest share of output units and have better precision (see Table 1).

Based on this observation, our primary goal is to optimize further the number of tokens copied from the TM. To do so, we propose simultaneously editing multiple fuzzy matches retrieved from memory, using a computational architecture – Multi-LevT,

---

[1] Copies and generations are directly infered from the sequence of edit operations used to compute the output sentence.

or $\texttt{TM}^N\texttt{-LevT}$ for short – which extends $\texttt{TM-LevT}$ to handle several initial translations. The benefit is twofold: (a) an increase in translation accuracy; (b) more transparency in the translation process. Extending $\texttt{TM-LevT}$ to $\texttt{TM}^N\texttt{-LevT}$ however requires solving multiple algorithmic and computational challenges related to the need to compute Multiple String Alignments (MSAs) between the matches and the reference translation, which is a notoriously difficult problem; and designing appropriate training procedures for this alignment module.

Our main contributions are the following:

1. a new variant of the $\texttt{LevT}$ model that explicitly maximizes target coverage (§4.2);

2. a new training regime to handle an extended set of editing operations (§3.3);

3. two novel multiway alignment (§4.2) and re-alignment (§6.2) algorithms;

4. experiments in 11 domains where we observe an increase of BLEU scores, COMET scores, and the proportion of copied tokens (§6).

Our code and experimental configurations are available on github.[2]

## 2 Preliminaries / Background

### 2.1 TM-based machine translation

Translation Memories, storing examples of past translations, is a primary component of professional Computer Assisted Translation (CAT) environments (Bowker, 2002). Given a translation request for source sentence $\mathbf{x}$, TM-based translation is a two-step process: (a) retrieval of one or several instances $(\tilde{\mathbf{x}}, \tilde{\mathbf{y}})$ whose source side resembles $\mathbf{x}$, (b) adaptation of retrieved example(s) to produce a translation. In this work, we mainly focus on step (b), and assume that the retrieval part is based on a fixed similarity measure $\Delta$ between $\mathbf{x}$ and stored examples. In our experiments, we use:

$$\Delta(\mathbf{x}, \tilde{\mathbf{x}}) = 1 - \frac{\text{ED}(\mathbf{x}, \tilde{\mathbf{x}})}{\max(|\mathbf{x}|, |\tilde{\mathbf{x}}|)}, \qquad (1)$$

with $\text{ED}(\mathbf{x}, \tilde{\mathbf{x}})$ the edit distance between $\mathbf{x}$ and $\tilde{\mathbf{x}}$ and $|\mathbf{x}|$ the length of $\mathbf{x}$. We only consider TM matches for which $\Delta$ exceeds a predefined threshold $\tau$ and filter out the remaining ones. The next step, adaptation, is performed by humans with CAT

[2]https://github.com/Maxwell1447/fairseq/

tools. Here, we instead explore ways to perform this step automatically, as in Example-Based MT (Nagao, 1984; Somers, 1999; Carl et al., 2004).

### 2.2 Adapting fuzzy matches with LevT

The Levenshtein transformer of Gu et al. (2019) is an encoder-decoder model which, given a source sentence, predicts edits that are applied to an initial translation in order to generate a revised output (Figure 1). The initial translation can either be empty or correspond to a match from a TM. Two editing operations – insertion and deletion – are considered. The former is composed of two steps: first, *placeholder insertion*, which predicts the position and number of new tokens; second, the *predictions of tokens* to fill these positions. Editing operations are applied iteratively in rounds of refinement steps until a final translation is obtained.

In $\texttt{LevT}$, these predictions rely on a joint encoding of the source and the current target and apply in parallel for all positions, which makes $\texttt{LevT}$ a representative of non-autoregressive translation (NAT) models. As editing operations are not observed in the training data, $\texttt{LevT}$ resorts to Imitation Learning, based on the generation of decoding configurations for which the optimal prediction is easy to compute. Details are in (Gu et al., 2019), see also (Xu and Carpuat, 2021), which extends it with a repositioning operation and uses it to decode with terminology constraints, as well as the studies of Niwa et al. (2022) and Xu et al. (2023) who also explore the use of $\texttt{LevT}$ in conjunction with TMs.

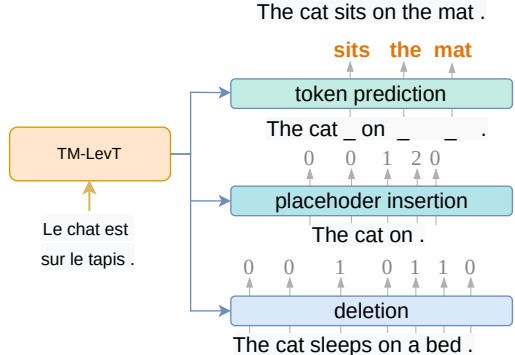

Figure 1: First decoding pass of $\texttt{TM-LevT}$, a variant of $\texttt{LevT}$ augmented with Translation Memories.

### 2.3 Processing multiple fuzzy matches

One of the core differences between $\texttt{TM}^N\texttt{-LevT}$ and $\texttt{LevT}$ is its ability to handle multiple matches. This

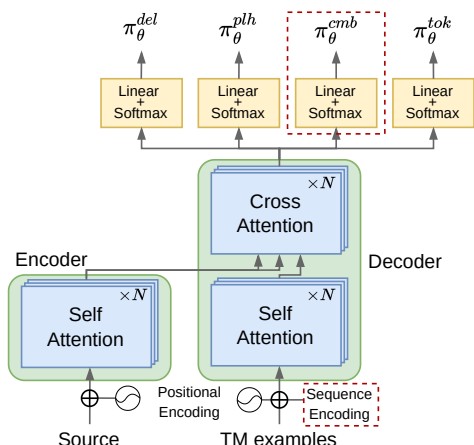

Figure 2: A high-level overview of TM$^N$–LevT's architecture. Additions w.r.t. TM–LevT are in a dashed box.

implies adapting the edit steps (in inference) and the roll-in policy (in imitation learning).

**Inference in TM$^N$-LevT**  Decoding follows the same key ideas as for LevT (see Figure 1) but enables co-editing an arbitrary number $N$ of sentences. Our implementation (1) applies deletion, then placeholder insertion simultaneously on each retrieved example; (2) combines position-wise all examples into one single candidate sentence; (3) performs additional steps as in LevT: this includes first completing *token prediction*, then performing *Iterative Refinement* operations that edit the sentence to correct mistakes and improve it (§3.2).

**Training in TM$^N$-LevT**  TM$^N$–LevT is trained with imitation learning and needs to learn the edit steps described above for both the first pass (1–2) and the iterative refinement steps (3). This means that we teach the model to perform the sequence of *correct* edit operations needed to iteratively generate the reference output, based on the step-by-step reproduction of what an expert 'teacher' would do. For this, we need to compute the *optimal operation* associated with each configuration (or state) (§4). The *roll-in* and *roll-out policies* specify how the model is trained (§3.3).

## 3   Multi-Levenshtein Transformer

### 3.1   Global architecture

TM$^N$–LevT has two modes of operations: (a) the combination of multiple TM matches into one single sequence through alignment, (b) the iterative refinement of the resulting sequence. In step (a), we use the Transformer encoder-decoder architec-

ture, extended with additional embedding and linear layers (see Figure 2) to accommodate multiple matches. In each of the $N$ retrieved instances $\mathbf{y} = (\mathbf{y}_1, \cdots, \mathbf{y}_N)$, $\mathbf{y}_{n,i}$ (the $i^{\text{th}}$ token in the $i^{\text{th}}$ instance) is encoded as $E_{\mathbf{y}_{n,i}} + P_i + S_n$, where $E \in \mathbb{R}^{|\mathcal{V}| \times d_{model}}$, $P \in \mathbb{R}^{L_{max} \times d_{model}}$ and $S \in \mathbb{R}^{(N+1) \times d_{model}}$ are respectively the token, position and sequence embeddings. The sequence embedding identifies TM matches, and the positional encodings are reset for each $\mathbf{y}_n$. The extra row in $S$ is used to identify the results of the combination and will yield a different representation for these single sequences.[3] Once embedded, TM matches are concatenated and passed through multiple Transformer blocks, until reaching the last layer, which outputs $(h_1, \cdots, h_{|\mathbf{y}|})$ for a single input match or $(h_{1,1}, \cdots, h_{1,|\mathbf{y}_1|}, \cdots, h_{N,1}, \cdots, h_{N,|\mathbf{y}_N|})$ in the case of multiple ones. The *learned policy* $\pi_\theta$ computes its decisions from these hidden states. We use four classifiers, one for each sub-policy:

1. *deletion*: predicts *keep* or *delete* for each token $\mathbf{y}_{n,i}^{del}$ with a projection matrix $A \in \mathbb{R}^{2 \times d_{model}}$:

$$\pi_\theta^{del}(d|n, i, \mathbf{y}_1^{del}, \cdots, \mathbf{y}_N^{del}; \mathbf{x}) = \text{softmax}\left(h_{n,i} A^T\right)$$

2. *insertion*: predicts the number of placeholder insertions between $\mathbf{y}_{n,i}^{plh}$ and $\mathbf{y}_{n,i+1}^{plh}$ with a projection matrix $B \in \mathbb{R}^{(K_{max}+1) \times 2d_{model}}$:

$$\pi_\theta^{plh}(p|n, i, \mathbf{y}_1^{plh}, \cdots, \mathbf{y}_N^{plh}; \mathbf{x}) = \text{softmax}\left([h_{n,i}, h_{n,i+1}] B^T\right),$$

with $K_{max}$ the max number of insertions.

3. *combination*: predicts if token $\mathbf{y}_{n,i}^{cmb}$ in sequence $n$ must be kept in the combination, with a projection matrix $C \in \mathbb{R}^{2 \times d_{model}}$:

$$\pi_\theta^{cmb}(c|n, i, \mathbf{y}_1^{cmb}, \cdots, \mathbf{y}_N^{cmb}; \mathbf{x}) = \text{softmax}\left(h_{n,i} C^T\right).$$

4. *prediction*: predicts a token in vocabulary $\mathcal{V}$ at each placeholder position, with a projection matrix $D \in \mathbb{R}^{|\mathcal{V}| \times d_{model}}$:

$$\pi_\theta^{tok}(t|n, i, \mathbf{y}^{tok}; \mathbf{x}) = \text{softmax}\left(h_j D^T\right)$$

Except for step 3, these classifiers are similar to those used in the original LevT.

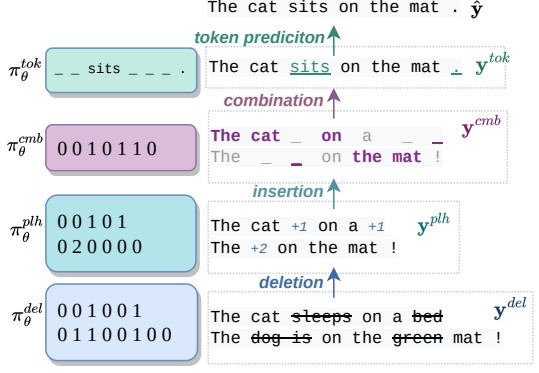

Figure 3: The first decoding pass in $\mathsf{TM}^N$-$\mathsf{LevT}$.

## 3.2 Decoding

Decoding is an iterative process: in a first pass, the $N$ fuzzy matches are combined to compute a candidate translation; then, as in $\mathsf{LevT}$, an additional series of *iterative refinement* rounds (Gu et al., 2019) is applied until convergence or timeout. Figure 3 illustrates the first pass, where $N = 2$ matches are first edited in parallel, then combined into one output.

To predict deletions (*resp.* insertions and token predictions), we apply the *argmax* operator to $\pi_\theta^{del}$ (*resp.* $\pi_\theta^{plh}$, $\pi_\theta^{tok}$). For combinations, we need to aggregate separate decisions $\pi_\theta^{cmb}$ (one per token and match) into one sequence. For this, at each position, we pick the most likely token.

During iterative refinement, we bias the model towards generating longer sentences since $\mathsf{LevT}$ outputs tend to be too short (Gu et al., 2019). As in $\mathsf{LevT}$, we add a penalty to the probability of inserting 0 placeholder in $\pi_\theta^{plh}$ (Stern et al., 2019). This only applies in the refinement steps to avoid creating more misalignments (see §6.2).

## 3.3 Imitation learning

We train $\mathsf{TM}^N$-$\mathsf{LevT}$ with Imitation Learning (Daumé et al., 2009; Ross et al., 2011), teaching the system to perform the right edit operation for each decoding state. As these operations are unobserved in the training data, the standard approach is to simulate decoding states via a *roll-in policy*; for each of these, the optimal decision is computed via an *expert policy* $\pi_*$, composed of intermediate experts $\pi_*^{del}$, $\pi_*^{plh}$, $\pi_*^{cmb}$, $\pi_*^{tok}$. The notion of optimality is discussed in §4. Samples of pairs (*state*, *decision*) are then used to train the system policy $\pi_\theta$.

First, from the initial set of sentences $\mathbf{y}^{init}$, the unrolling of $\pi_*$ produces intermediate states $(\mathbf{y}^{del}, del^*)$, $(\mathbf{y}^{plh}, plh^*)$, $(\mathbf{y}^{cmb}, cmb^*)$, $(\mathbf{y}^{tok}, tok^*)$ (see top left in Figure 4). Moreover, in this framework, it is critical to mitigate the *exposure bias* and generate states that result from non-optimal past decisions (Zheng et al., 2023). For each training sample $(\mathbf{x}, \mathbf{y}_1, \cdots, \mathbf{y}_N, \mathbf{y}_*)$, we simulate multiple additional states as follows (see Figure 4 for the full picture). We begin with the operations involved in the first decoding pass:[4]

1. **Additional triplets[♯]**: $\pi^{rnd \cdot del \cdot N}$ turns $\mathbf{y}_*$ into $N$ random substrings, which simulates the edition of $N$ artificial examples.

2. **Token selection[♯]** (uses $\pi^{sel}$): our expert policy never aligns two distinct tokens at a given position (§4.3). We simulate such cases that may occur at inference, as follows: with probability $\gamma$, each <PLH> is replaced with a random token from fuzzy matches (Figure 5).

The expert always completes its translation in one decoding pass. Policies used in iterative refinement are thus trained with the following simulated states, based on roll-in and roll-out policies used in $\mathsf{LevT}$ and its variants (Gu et al., 2019; Xu et al., 2023; Zheng et al., 2023):

3. **Add missing words** (uses $\pi^{rnd \cdot del \cdot 1}$): with probability $\alpha$, $\mathbf{y}^{post \cdot plh} = \mathbf{y}_*$. With probability $1 - \alpha$, generate a subsequence $\mathbf{y}^{post \cdot plh}$ with length sampled uniformly in $[0, |\mathbf{y}_*|]$.

4. **Correct mistakes** (uses $\pi_\theta^{tok}$): using the output of token prediction $\mathbf{y}^{post \cdot del}$, teach the model to erase the wrongly predicted tokens.

5. **Remove extra tokens[♯]** (uses $\pi_\theta^{ins}$, $\pi_\theta^{tok}$): insert placeholders in $\mathbf{y}^{post \cdot tok}$ and predict tokens, yielding $\mathbf{y}^{post \cdot del \cdot extra}$, which trains the model to delete wrong tokens. These sequences differ from case (4) in the way <PLH> are inserted.

6. **Predict token[♯]** (uses $\pi^{rnd \cdot msk}$): each token in $\mathbf{y}_*$ is replaced by <PLH> with probability $\varepsilon$. As token prediction applies for both decoding steps, these states also improve the first pass.

The expert decisions (e.g. inserting deleted tokens like in state (3) ; or deleting wrongly predicted

---

[3] $\mathbf{y}^{op}$ denotes an intermediary sequence before applying edit operation *op*. $\mathbf{y}_n^{op} \in \mathbb{N}^L$ is encoded with $S_n$; $\mathbf{y}^{op} \in \mathbb{N}^L$ with $S_N$.

[4] Families of (*state*, *decision*) pairs that are novel with respect to $\mathsf{TM}$-$\mathsf{LevT}$ are marked with [♯].

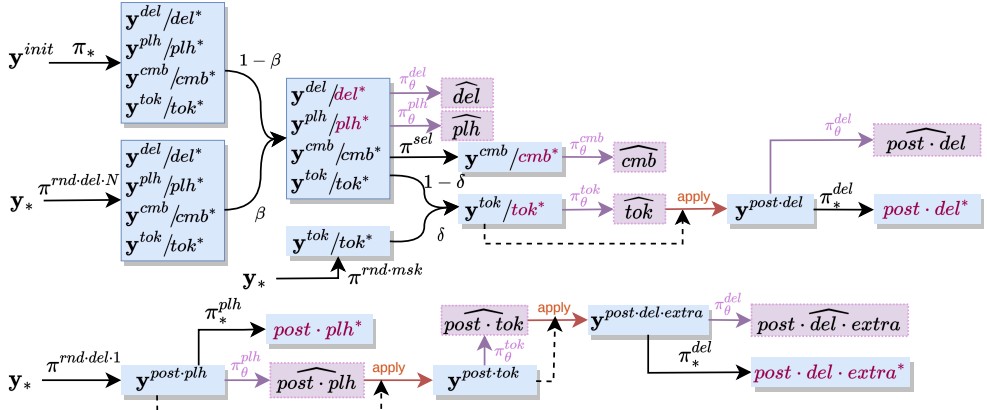

Figure 4: Roll-in policies used in training. Blue cells contain sets of target sentences (e.g. $\mathbf{y}^{del}$), optionally associated with the expert prediction (e.g. $del^*$). Model's predictions are in Thistle and circumflexed (e.g. $\widehat{del}$). Pairs of model / expert predictions are summed in the loss: $(\widehat{del}, del^*)$, $(\widehat{plh}, plh^*)$, $(\widehat{cmb}, cmb^*)$, $(\widehat{post \cdot del}, post \cdot del^*)$, $(\widehat{post \cdot plh}, post \cdot plh^*)$, $(\widehat{post \cdot del \cdot extra}, post \cdot plh \cdot extra^*)$. "post" denotes policies applied in refinement steps.

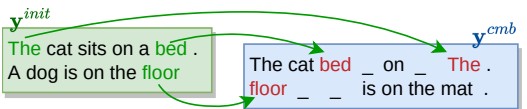

Figure 5: Noising $\mathbf{y}^{cmb}$ with $\pi^{sel}$ using tokens from $\mathbf{y}^{init} = (\mathbf{y}_1, \cdots, \mathbf{y}_N)$.

tokens in state (4)) associated with most states are obvious, except for the initial state and state (5), which require an optimal alignment computation.

## 4 Optimal Alignment

Training the combination operation introduced above requires specifying the expert decision for each state. While LevT derives its expert policy $\pi_*$ from the computation of *edit distances*, we introduce another formulation based on the computation of *maximal covers*. For $N$=1, these formulations can be made equivalent[5] (Gusfield, 1997).

### 4.1 N-way alignments

We formulate the problem of optimal editing as an N-way alignment problem (see figure 6) that we define as follows. Given $N$ examples $(\mathbf{y}_1, \cdots, \mathbf{y}_N)$ and the target sentence $\mathbf{y}_*$, a N-way alignment of $(\mathbf{y}_1, \cdots, \mathbf{y}_N)$ w.r.t. $\mathbf{y}_*$ is represented as a bipartite graph $(V, V_*, E)$, where $V$ is further partitioned into $N$ mutually disjoint subsets $V_1 \ldots V_N$. Vertices in each $V_n$ (resp. $V_*$) correspond to tokens in $\mathbf{y}_n$ (resp. $\mathbf{y}_*$). Edges $(n, i, j) \in E$ connect node $i$

in $V_n$ to node $j$ in $V_*$. An N-way alignment satisfies properties (i)-(ii):

(i) Edges connect identical (matching) tokens:
$(n, i, j) \in E \Rightarrow \mathbf{y}_{n,i} = \mathbf{y}_{*,j}$.

(ii) Edges that are incident to the same subset $V_n$ do not cross:
$(n, i, j), (n, i', j') \in E \Rightarrow (i'-i)(j'-j) > 0$.

An optimal N-way alignment $E_*$ **maximizes the coverage** of tokens in $\mathbf{y}_*$, then **the total number of edges**, where $\mathbf{y}_{*,j}$ is covered if there exists at least one edge $(n, i, j) \in E$. Denoting $\mathbb{E}$ the set of alignments maximizing target coverage:

$$\mathbb{E} = \arg \max_E |\{\mathbf{y}_{*,j} : \exists (n, i), (n, i, j) \in E\}|.$$
$$E_* = \arg \max_{E \in \mathbb{E}} |E|.$$

### 4.2 Solving optimal alignment

Computing the optimal N-way alignment is NP-hard (see Appendix D). This problem can be solved using Dynamic Programming (DP) techniques similar to Multiple Sequence Alignment (MSA) (Carrillo and Lipman, 1988) with a complexity $O(N|\mathbf{y}_*| \prod_n |\mathbf{y}_n|)$. We instead implemented the following two-step heuristic approach:

1. separately compute alignment graphs between each $\mathbf{y}_n$ and $\mathbf{y}_*$, then extract $k$-best 1-way alignments $\{E_{n,1} \ldots E_{n,k}\}$. This requires time $O(k|\mathbf{y}_n||\mathbf{y}_*|)$ using DP (Gusfield, 1997);

---

[5]When the cost of replace is higher than insertion + deletion. This is the case in the original LevT code.

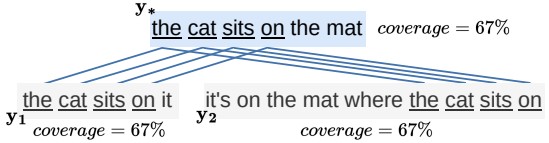 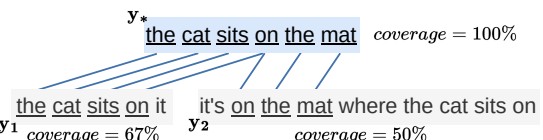

(a) Combination of the optimal 1-way alignments.

(b) Optimal 2-way alignment.

Figure 6: Illustration of the optimal N-way alignment which maximizes a global coverage criterion (6b), while independent alignments do not guarantee optimal usage of information present in TM examples (6a).

2. search for the optimal recombination of these graphs, selecting 1-way alignments $(E_{1,k_1} \ldots E_{N,k_N})$ to form $E_* = \bigcup_n E_{n,k_n}$. Assuming $N$ and $k$ are small, we perform an exhaustive search in $O(k^N)$.

### 4.3 From alignments to edits

From an alignment $(V, V_*, E)$, we derive the optimal edits needed to compute $\mathbf{y}_*$, and the associated intermediary sequences. Edges in $E$ indicate the tokens that are preserved throughout this process:

1. *deletion*: $\forall n, \forall i, \mathbf{y}_{n,i}$ is kept only if $(n, i, j) \in E$ for some $j$; otherwise it is deleted. The resulting sequences are $\{\mathbf{y}_n^{plh}\}_{n=1\ldots N}$.

2. *insertion*: Placeholders are inserted between successive tokens in all $\mathbf{y}_n^{plh}$, resulting in the set $\{\mathbf{y}_n^{cmb}\}_{n=1\ldots N}$, under the constraints that (a) all $\mathbf{y}_n^{cmb}$ have the same length as $\mathbf{y}_*$ and (b) non-placeholder tokens $\mathbf{y}_{n,i}^{cmb}$ are equal to the reference token $\mathbf{y}_{*,i}$.

3. *combination*: Sequences $\{\mathbf{y}_n^{cmb}\}_{n=1\ldots N}$ are combined into $\mathbf{y}^{tok}$ such that for each position $i$, $\mathbf{y}_{n,i}^{cmb} \neq \texttt{<PLH>} \Rightarrow \mathbf{y}_i^{tok} = \mathbf{y}_{n,i}^{cmb}$. If $\forall n, \mathbf{y}_{n,i}^{cmb} = \texttt{<PLH>}$, then $\mathbf{y}_i^{tok} = \texttt{<PLH>}$.

4. *prediction*: The remaining $\texttt{<PLH>}$ symbols in $\mathbf{y}^{tok}$ are replaced by the corresponding target token in $\mathbf{y}_*$ at the same position.

The *expert policy* $\pi_*$ edits examples $\mathbf{y}_1, \cdots, \mathbf{y}_N$ into $\mathbf{y}_*$ based on the optimal alignment $(V, V_*, E_*)$. It comprises $\pi_*^{del}$, $\pi_*^{plh}$, $\pi_*^{cmb}$, and $\pi_*^{tok}$, corresponding to the four steps listed above.

## 5 Experiments

### 5.1 Data and metrics

We focus on translation from English to French and consider multiple domains. This allows us to consider a wide range of scenarios, with a varying density of matching examples: our datasets include

ECB, EMEA, Europarl, GNOME, JRC-Acquis, KDE4, PHP, Ubuntu, where high-quality matches are often available, but also News-Commentary, TED2013, and Wikipedia, where matches are more scarce (see Table 6, §B).

For each training sample $(\mathbf{x}, \mathbf{y})$, we retrieve up to 3 *in-domain* matches. We filter matches $\mathbf{x}_n$ to keep only those with $\Delta(\mathbf{x}, \mathbf{x}_n) > 0.4$. We then manually split each of the 11 datasets into *train*, *valid*, *test-0.4*, *test-0.6*, where the *valid* and *test* sets contain 1,000 lines each. *test-0.4* (resp. *test-0.6*) contains samples whose best match is in the range $[0.4, 0.6[$ (resp. $[0.6, 1[$). As these two test sets are only defined based on the *best match score*, it may happen that some test instances will only retrieve 1 or 2 close matches (statistics are in Table 6).

For the pre-training experiments (§6.2), we use a subsample of 2M random sentences from WMT'14. For all data, we use Moses tokenizer and 32k BPEs trained on WMT'14 with SentencePiece (Kudo, 2018). We report BLEU scores (Papineni et al., 2002) and ChrF scores (Popović, 2015) as computed by *SacreBLEU* (Post, 2018) and COMET scores (Rei et al., 2020).

### 5.2 Architecture and settings

Our code[6] extends Fairseq[7] implementation of LevT in many ways. It uses Transformer models (Vaswani et al., 2017) (parameters in Appendix A). Roll-in policy parameters (§3.3) are empirically set as: $\alpha$=0.3, $\beta$=0.2, $\gamma$=0.2, $\delta$=0.2, $\varepsilon$=0.4. The AR baseline uses OpenNMT (Klein et al., 2017) and uses the same data as TM-LevT (Appendix A).

## 6 Results

### 6.1 The benefits of multiple matches

We compare two models in Table 2: one trained with one TM match, the other with three. Each

---

[6]https://github.com/Maxwell1447/fairseq
[7]https://github.com/facebookresearch/fairseq

| Model \N | 1 | 2 | 3 | all |
|---|---|---|---|---|
| size | 4,719 | 2,369 | 14,912 | 22,000 |
| TM$^1$-LevT | 45.8/63.6 | 48.7/65.0 | 55.0/68.4 | 52.0/66.8 |
| | 19.6 | 26.2 | 41.5 | 35.0 |
| TM$^3$-LevT | 46.6/64.1 | 50.0/65.8 | 56.0/69.3 | 53.0/67.5 |
| | 14.0 | 16.0 | 38.2 | 30.8 |

Table 2: BLEU/ChrF and COMET scores on the full test set. All BLEU/ChrF differences are significant ($p = 0.05$).

model is evaluated with, at most, the same number of matches seen in training. This means that TM$^1$-LevT only uses the 1-best match, even when more examples are found. In this table, test sets *test-0.4* and *test-0.6* are concatenated, then partitioned between samples for which exactly 1, 2, and 3 matches are retrieved. We observe that TM$^3$-LevT, trained with 3 examples, consistently achieves better BLEU and ChrF scores than TM$^1$-LevT, even in the case $N$=1, where we only edit the closest match.[8] These better BLEU scores are associated with a larger number of copies from the retrieved instances, which was our main goal (Table 3). Similar results for the other direction are reported in the appendix § E (Table 7).

| | | TM-LevT | TM$^1$-LevT | TM$^3$-LevT |
|---|---|---|---|---|
| unigram | *copy* | 64.9 | 64.5 | 68.8 |
| | *gen* | 35.1 | 35.5 | 31.2 |
| bigram | *copy-copy* | 55.0 | 54.5 | 58.0 |
| | *copy-gen* | 8.9 | 9.0 | 10.1 |
| | *gen-copy* | 10.7 | 10.8 | 11.0 |
| | *gen-gen* | 25.4 | 25.7 | 20.9 |

Table 3: Proportion of unigrams and bigram from a given origin (copy vs. generation) for various models.

We report the performance of systems trained using $N$=1, 2, 3 for each domain and test set in Table 4 (BLEU) and 12 (COMET). We see comparable average BLEU scores for $N$=1 and $N$=3, with large variations across domains, from which we conclude that: (a) using 3 examples has a smaller return when the best match is poor, meaning that bad matches are less likely to help (*test-0.4* vs. *test-0.6*); (b) using 3 examples seems advantageous for narrow domains, where training actually exploits several close matches (see also Appendix F). We finally note that COMET scores[9] for TM$^3$-LevT are

---

[8]This is because the former model has been fed with more examples during training, which may help regularization.

[9]Those numbers are harder to interpret, given the wide range of COMET scores across domains (from ≈ -40 to +86).

always slightly lower than for TM$^1$-LevT, which prompted us to develop several extensions.

## 6.2 Improving TM$^N$-LevT

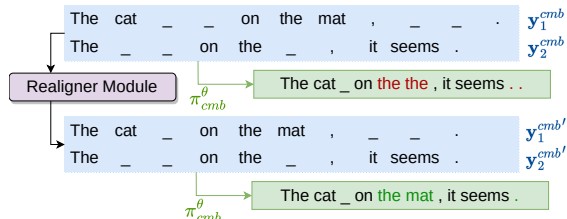

Figure 7: Fixing misalignments with realignment

**Realignment** In preliminary experiments, we observed that small placeholder prediction errors in the first decoding pass could turn into catastrophic misalignments (Figure 7). To mitigate such cases, we introduce an additional realignment step during inference, where some predicted placeholders are added/removed if this improves the global alignment. Realignment is formulated as an optimization problem aimed to perform a tradeoff between the score $-\log \pi_\theta^{plh}$ of placeholder insertion and an alignment cost (see Appendix C).

We assess realignment for $N$=3 (Tables 4 and 12) and observe small, yet consistent average gains (+0.2 BLEU, +1.5 COMET) for both test sets.

**Pre-training** Another improvement uses pre-training with synthetic data. For each source/target pair $(\mathbf{x}, \mathbf{y})$ in the pre-training corpus, we simulate $N$ fuzzy matches by extracting from $\mathbf{y}$ $N$ substrings $\mathbf{y}_n$ of length $\approx |\mathbf{y}| \cdot r$, with $r \in [0, 1]$. Each $\mathbf{y}_n$ is then augmented as follows:

1. We randomly insert placeholders to increase the length by a random factor between 1 and $1 + f$, $f = 0.5$ in our experiments.

2. We use the CamemBERT language model (Martin et al., 2020) to fill the masked tokens.

These artificial instances simulate diverse fuzzy matches and are used to pre-train a model, using the same architecture and setup as in §5.2. Pre-training yields markedly higher scores than the baseline (+1.3 BLEU, +6.4 COMET for *test-0.4* and +0.9 BLEU, +4.6 COMET for *test-0.6*). Training curves also suggest that pre-trained models are faster to converge. Combining with realignment yields additional gains for TM$^3$-LevT, which *outperforms* TM$^1$-LevT *in all domains and both metrics*.

| | | ECB | EME | Epp | GNO | JRC | KDE | News | PHP | TED | Ubu | Wiki | **all** |
|---|---|---|---|---|---|---|---|---|---|---|---|---|---|
| *test-0.4* | AR | **63.0** | **63.6** | **43.6** | **69.3** | **75.1** | **62.8** | **28.8** | **41.2** | **42.2** | **59.1** | **42.2** | **55.8** |
| | TM-LevT | *50.5* | *50.7* | *31.3* | *54.3* | *62.4* | *47.9* | *18.0* | *30.1* | *24.2* | *43.3* | *29.8* | *42.8* |
| | TM¹-LevT | 53.1 | 53.7 | 35.5 | 60.3 | 65.6 | 51.8 | 22.2 | 31.7 | 30.2 | 48.8 | 32.0 | 46.2 |
| | TM²-LevT | **54.0** | 54.3 | *34.0* | 60.5 | 66.0 | 53.2 | 20.7 | **33.7** | 28.9 | 48.0 | 32.6 | 46.5 |
| | TM³-LevT | 53.9 | **55.6** | *34.2* | 60.8 | 66.0 | 53.5 | 20.4 | 33.1 | 28.6 | 47.5 | 32.9 | **46.5** |
| | +pre-train | **54.9** | **55.9** | 34.4 | **62.7** | 67.4 | 54.1 | *21.1* | **34.7** | 30.1 | 49.3 | **33.5** | **47.5** |
| | +realign | **54.4** | **55.9** | 34.4 | **61.2** | 66.2 | 53.2 | 20.4 | 33.3 | 28.4 | 47.9 | **33.1** | 46.7 |
| | +both | **55.0** | **56.0** | 34.9 | **62.8** | 67.5 | 54.0 | *21.4* | **34.8** | 30.8 | 49.6 | **33.9** | **47.8** |
| *test-0.6* | AR | **69.7** | **70.4** | **57.4** | **80.6** | **82.4** | **68.2** | **26.1** | **46.4** | **62.5** | **68.5** | **68.7** | **66.6** |
| | TM-LevT | *59.0* | *64.0* | *45.8* | *66.9* | *73.5* | *53.4* | *18.8* | *34.7* | *49.1* | *53.2* | *58.9* | *55.8* |
| | TM¹-LevT | 60.5 | 64.6 | 48.9 | 69.7 | 75.7 | 57.2 | 21.0 | 36.2 | 55.0 | 58.3 | 62.2 | 58.2 |
| | TM²-LevT | **62.7** | **67.0** | **50.0** | **71.7** | 76.2 | **60.2** | 21.7 | **38.6** | 54.2 | **59.8** | 62.8 | **59.7** |
| | TM³-LevT | **63.8** | **67.4** | **50.0** | **71.1** | 76.4 | **60.0** | 21.5 | **39.2** | 54.3 | **59.6** | 62.3 | **60.0** |
| | +pre-train | **64.9** | **68.3** | 50.3 | **72.7** | 77.3 | **62.3** | 21.8 | **40.7** | 54.6 | **61.3** | **65.0** | **61.1** |
| | +realign | **64.0** | **68.0** | 50.2 | **71.5** | 76.5 | 59.9 | 21.6 | **39.0** | 54.7 | **60.0** | 63.1 | **60.2** |
| | +both | **65.0** | **68.3** | 50.8 | **73.7** | 77.4 | **62.3** | 22.0 | **40.6** | 54.7 | **61.4** | **65.3** | **61.3** |

Table 4: Per domain BLEU scores for TM-LevT, TM$^N$-LevT and variants. Bold (resp. italic) for scores significantly higher (resp. lower) than TM¹-LevT ($p = 0.05$). $p$-values from SacreBLEU paired bootstrap resampling ($n = 1000$). The Autoregressive (AR) system is our implementation of (Bulte and Tezcan, 2019).

**Knowledge distillation** *Knowledge Distillation* (KD) (Kim and Rush, 2016) is used to mitigate the effect of multimodality of NAT models (Zhou et al., 2020) and to ease the learning process. We trained a TM$^N$-LevT model with distilled samples $(\mathbf{x}, \tilde{\mathbf{y}}_1, \cdots, \tilde{\mathbf{y}}_N, \tilde{\mathbf{y}})$, where automatic translations $\tilde{\mathbf{y}}_i$ and $\tilde{\mathbf{y}}$ are derived from their respective source $\mathbf{x}_i$ and $\mathbf{x}$ with an auto-regressive teacher trained with a concatenation of all the training data.

We observe that KD is beneficial (+0.3 BLEU) for low-scoring matches (*test-0.4*) but hurts performance (-1.7 BLEU) for the better ones in *test-0.6*. This may be because the teacher model, with a BLEU score of 56.7 on the *test-0.6*, fails to provide the excellent starting translations the model can access when using non-distilled data.

### 6.3 Ablation study

We evaluate the impact of the various elements in the mixture roll-in policy via an ablation study (Table 13). Except for $\pi^{sel}$, every new element in the roll-in policy increases performance. As for $\pi^{sel}$, our system seems to be slightly better with than without. An explanation is that, in case of misalignment, the model is biased towards selecting the first, most similar example sentence. As an ablation, instead of aligning by globally maximizing coverage (§ 4.2), we also compute alignments that maximize coverage independently as in figure 6a.

A complete run of TM$^N$-LevT is in Appendix F.

| | test-0.4 | test-0.6 |
|---|---|---|
| TM³-LevT | **46.5** | **60.1** |
| -sel | 46.2 | 60.0 |
| -delx | 44.8 | 58.6 |
| -rnd-del | 38.6 | 51.9 |
| -rnd-mask | 46.0 | 59.0 |
| -dum-plh | 41.0 | 50.9 |
| -indep-align | 42.6 | 56.4 |

Table 5: Ablation study. We build models with variable roll-in policies: -sel: no random selection noise ($\gamma$=0); -delx: no extra deletion; -rd-del: no random deletion ($\beta$=0); -mask: no random mask ($\delta$=0); -dum-plh: never start with $\mathbf{y}_{post \cdot del}=\mathbf{y}_*$ ($\alpha$=0); -indep-align: alignments are independent. Full results in Appendix F.

## 7 Related Work

As for other Machine Learning applications, such as text generation (Guu et al., 2018), efforts to integrate a retrieval component in neural-based MT have intensified in recent years. One motivation is to increase the transparency of ML models by providing users with tangible traces of their internal computations in the form of retrieved examples (Rudin, 2019). For MT, this is achieved by integrating fuzzy matches retrieved from memory as an additional conditioning context. This can be performed simply by concatenating the retrieved target instance to the source text (Bulte and Tezcan, 2019), an approach that straightforwardly accommodates several TM matches (Xu et al., 2020), or the simultaneous exploitation of their source and target sides (Pham et al., 2020). More complex schemes to combine retrieved examples with the source sentence are in (Gu et al., 2018; Xia et al.,

2019; He et al., 2021b). The recent work of Cheng et al. (2022) handles multiple complementary TM examples retrieved in a *contrastive manner* that aims to enhance source coverage. Cai et al. (2021) also handle multiple matches and introduce two novelties: (a) retrieval is performed in the target language and (b) similarity scores are trainable, which allows to evaluate retrieved instances based on their usefulness in translation. Most of these attempts rely on an auto-regressive (AR) decoder, meaning that the impact of TM match(es) on the final output is only indirect.

The use of TM memory match with a NAT decoder is studied in (Niwa et al., 2022; Xu et al., 2023; Zheng et al., 2023), which adapt LevT for this specific setting, using one single retrieved instance to initialize the edit-based decoder. Other evolutions of LevT, notably in the context of constraint decoding, are in (Susanto et al., 2020; Xu and Carpuat, 2021), while a more general account of NAT systems is in (Xiao et al., 2023).

Zhang et al. (2018) explore a different set of techniques to improve translation using retrieved segments instead of full sentences. Extending KNN-based language models (He et al., 2021a) to the conditional case, Khandelwal et al. (2021) proposes $k$-nearest neighbor MT by searching for target tokens that have similar contextualized representations at each decoding step, an approach further elaborated by Zheng et al. (2021); Meng et al. (2022) and extended to chunks by Martins et al. (2022).

## 8 Conclusion and Outlook

In this work, we have extended the Levenshtein Transformer with a new combination operation, making it able to simultaneously edit multiple fuzzy matches and merge them into an initial translation that is then refined. Owing to multiple algorithmic contributions and improved training schemes, we have been able to (a) increase the number of output tokens that are copied from retrieved examples; (b) obtain performance improvements compared to using one single match. We have also argued that retrieval-based NMT was a simple way to make the process more transparent for end users.

Next, we would like to work on the retrieval side of the model: first, to increase the diversity of fuzzy matches e.g. thanks to contrastive retrieval, but also to study ways to train the retrieval mechanism and extend this approach to search monolingual (target side) corpora. Another line of work will combine our techniques with other approaches to TM-based NMT, such as keeping track of the initial translation(s) on the encoder side.

## 9 Limitations

As this work was primarily designed a feasibility study, we have left aside several issues related to performance, which may explain the remaining gap with published results on similar datasets. First, we have restricted the encoder to only encode the source sentence, even though enriching the input side with the initial target(s) has often been found to increase performance (Bulte and Tezcan, 2019), also for NAT systems (Xu et al., 2023). It is also likely that increasing the number of training epochs would yield higher absolute scores (see Appendix F).

These choices were made for the sake of efficiency, as our training already had to fare with the extra computing costs incurred by the alignment procedure required to learn the expert policy. Note that in comparison, the extra cost of the realignment procedure is much smaller, as it is only paid during inference and can be parallelized on GPUs.

We would also like to outline that our systems do not match the performance of an equivalent AR decoder, a gap that remains for many NAT systems (Xiao et al., 2023). Finally, we have only reported here results for one language pair – favoring here domain diversity over language diversity – and would need to confirm the observed improvements on other language pairs and conditions.

## 10 Acknowledgements

This work was performed using HPC resources from GENCI-IDRIS (Grant 2022-AD011013583) and Lab-IA from Saclay-IA.

The authors wish to thank Dr. Jitao Xu and Dr. Caio Corro for their guidance and help.

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

## A   Model Configuration

We use a Transformer architecture with embeddings of dimension 512; feed-forward layers of size 2048; number of heads 8; number of encoder and decoder layers: 6; batch size: 3000 tokens; shared-embeddings; dropout: 0.3; number of GPUs: 6. The maximal number of additional placeholders is $K_{max} = 64$.

During training, we use Adam optimizer with $(\beta_1, \beta_2)$=(0.9, 0.98); inverse sqrt scheduler; learning rate: $5e^{-4}$; label smoothing: 0.1; warmup updates: 10,000; float precision: 16. We fixed the number of iterations at 60k. For decoding, we use iterative refinement with an empty placeholder penalty of 3, and a max number of iterations of 10 (Gu et al., 2019).

For the $n$-way alignment (§4.1), we use $k$=10.

The hyper-parameters of the realigner (§C) were tuned on a subset of 1k samples extracted from the ECB training set.

Metrics are used with default settings: SacreBLEU signature is `nrefs:1|case:mixed|eff:no|tok:13a|smooth:exp|version:2.1.0`; the ChrF signature is `nrefs:1|case:mixed|eff:yes|nc:6|nw:0|space:no|version:2.1.0`; as for COMET we use the default model of version 1.1.3: `Unbabel/wmt22-comet-da`.

## B   Data Analysis

Table 6 contains statistics about all 11 domains. They notably highlight the relationship between the average number of retrieved sentences during training and the ability of TM$^3$-LevT to perform better than TM$^1$-LevT in Table 4. The domains with retrieval rates lesser than 1 (Epp, News, TED, Ubu) have quite a broad content, meaning that training instances have fewer close matches, which also means that for these domains, TM$^3$-LevT hardly sees two or three examples that it needs to use in inference.

## C   Realignment

The realignment process is an extra inference step aiming to improve the result of the placeholder insertion stage. To motivate our approach, let us consider the following sentences before placeholder insertion:

$$\mathbf{y}_0^{plh}: \quad < \quad A \quad B \quad C \quad > \quad \times$$
$$\mathbf{y}_1^{plh}: \quad < \quad B \quad C \quad > \quad \times \quad \times$$
$$\mathbf{y}_2^{plh}: \quad < \quad A \quad D \quad C \quad D \quad >,$$

where letters represent tokens, $\times$ denotes padding, $<$ and $>$ respectively stand for `<BOS>` and `<EOS>`.

The output of this stage is a prediction for all pairs of consecutive tokens. This prediction takes the form of a tensor $\log \pi_\theta^{plh}$ of dimensions $N \times (L-1) \times (K_{max}+1)$, corresponding respectively to the number of retrieved sentences $N$, the maximum sentence length $L$, and the maximum number of additional placeholders $K_{max}$.

Let $P$ (a $N \times (L-1)$ tensor) denote the $\arg\max$,

$$\text{e.g. } P = \begin{matrix} 0 & 0 & 0 & 2 & 0 \\ 0 & 0 & 1 & 0 & 0 \\ 0 & 1 & 0 & 0 & 0 \end{matrix}$$

Inserting the prescribed number of placeholders (figured by _) then yields the following $\mathbf{y}^{cmb}$:

$$\mathbf{y}_0^{cmb}: \quad < \quad A \quad B \quad C \quad \_ \quad \_ \quad >$$
$$\mathbf{y}_1^{cmb}: \quad < \quad B \quad C \quad \_ \quad > \quad \times \quad \times$$
$$\mathbf{y}_2^{cmb}: \quad < \quad A \quad \_ \quad D \quad C \quad D \quad >$$

This result is far from perfect, as it fails to align the repeated occurrences of $C$. For instance, a preferable alignment requiring 3 changes (1 change consists in a modification of $\pm 1$ in $P$) could be:

$$\mathbf{y}_0^{cmb'}: \quad < \quad A \quad B \quad C \quad \_ \quad > \quad \times$$
$$\mathbf{y}_1^{cmb'}: \quad < \quad \_ \quad B \quad C \quad \_ \quad > \quad \times$$
$$\mathbf{y}_2^{cmb'}: \quad < \quad A \quad D \quad C \quad D \quad > \quad \times$$

The general goal of realignment is to improve such alignments by performing a small number of changes in $P$. We formalize this problem as a search for a good tradeoff between (a) the individual placeholder prediction scores, aggregated in $\mathcal{L}_L$ (likelihood loss) and (b) $\mathcal{L}_A$ an alignment loss. Under its simplest form, this problem is again an optimal multisequence alignment problem, for which exact dynamic programming solutions are computationally intractable in our setting.

We instead develop a continuous relaxation that can be solved effectively with SGD and is also easy to parallelize on GPUs. We, therefore, relax the integer condition for $P$ and assume that $P_{i,j}$ can take continuous real values in $[0, K_{max}]$, then solve the continuous optimization problem before turning $P_{i,j}$ values back into integers.

The likelihood loss aims to keep the $P_{i,j}$ values close to the model predictions. Denoting $(\mu, \sigma)$ respectively the mean and variance of the model

| domain | ECB | EME | Epp | GNO | JRC | KDE | News | PHP | TED | Ubu | Wiki | all |
|---|---|---|---|---|---|---|---|---|---|---|---|---|
| size | 195k | 373k | 2.0M | 55k | 503k | 180k | 151k | 16k | 159k | 9k | 803k | 4.4M |
| retrieval rate | 1.66 | 2.12 | 0.91 | 1.18 | 1.71 | 1.10 | 0.24 | 1.20 | 0.96 | 0.62 | 1.22 | 1.18 |
| mean length | 29.2 | 16.7 | 26.6 | 9.4 | 28.8 | 10.5 | 26.4 | 14.5 | 17.7 | 5.2 | 19.6 | 18.6 |

Table 6: Number of samples, average number of retrieved sentences and average length of sentences after tokenization for all 11 domains.

predictions, our initial version of this loss is

$$\mathcal{L}_L(P) = \sum_{i,j} \frac{(P_{i,j} - \mu_{i,j})^2}{2\sigma^2}.$$

In practice, we found that using a weighted average $\hat{\mu}$ and clamping the variance $\hat{\sigma}^2$ both yield better realignments, yielding:

$$\mathcal{L}_L(P) = \sum_{i,j} \frac{(P_{i,j} - \hat{\mu}_{i,j})^2}{2\hat{\sigma}^2}$$

To define the *alignment loss*, we introduce a *position matrix* $X$ of dimension $N \times L$ in $\mathbb{R}^+$, where $X_{n,i}$ corresponds to the (continuous)position of token $y_{n,i}$ after inserting a real number of placeholders. $X$ is defined as:

$$X_{n,i}(P) = i + \sum_{j<i} P_{n,j}$$

with $i$ the number of tokens occuring before $X_{n,i}$ and $\sum_{j<i} P_{n,j}$ the cumulated sum of placeholders. Using $X$, we derive the distance tensor $D$ of dimension $N \times L \times N \times L$ in $\mathbb{R}^+$ as:

$$D_{n,i,m,j}(P) = |X_{n,i} - X_{m,j}|$$

Finally, let $G$ be an $N \times L \times N \times L$ alignment graph tensor, where $G_{n,i,m,j} = 1$ if and only if $y_{n,i} = y_{m,j}$ and $n \neq m$ and $D_{n,i,m,j} < D_{max}$. $G$ connects identical tokens in different sentences when their distance after placeholder insertion is at most $D_{max}$. This last condition avoids perturbations from remote tokens that coincidentally appear to be identical.

Each token $y_{n,i}$ is associated with an individual loss:

$$d_{n,i}(P) = \begin{cases} \min_{m,j} \{D_{n,i,m,j}(P) : G_{n,i,m,j} = 1\} \\ \quad \text{if } \exists (m,j) \text{ s.t. } G_{n,i,m,j} = 1 \\ 0 \text{ otherwise.} \end{cases}$$

The *alignment loss* aggregates these values over sentences and positions as:

$$\mathcal{L}_A(P) = \sum_{n=0}^{N-1} \sum_{i=0}^{L-1} d_{n,i}(P)$$

A final ingredient in our realignment model is related to the final discretization step. To avoid rounding errors, we further constrain the optimization process to deliver near-integer solutions. For this, we also include a *integer constraint* loss defined as :

$$\mathcal{L}_{int}(P) = \mu_t \sum_{i,j} \sin^2(\pi P_{i,j})$$

where $\mu_t$ controls the scale of $\mathcal{L}_{int}(P)$. As $x \to \sin^2(\pi x)$ reaches its minimum 0 for integer values, minimizing $\mathcal{L}_{int}(P)$ has the effect of enforcing a near-integer constraint to our solutions. Overall, we minimize in $P$:

$$\mathcal{L} = \mathcal{L}_L(P) + \mathcal{L}_A(P) + \mathcal{L}_{int}(P),$$

slowly increasing the scale of $\mu_t$ according to the following schedule

$$\mu_t = \begin{cases} 0 & \text{if } t < t_0 \\ \mu_T & \text{if } t > T \\ \mu_T \frac{(t-t_0)^2}{(T-t_0)^2} & \text{otherwise} \end{cases},$$

with $t_0, T$ the timestamps for respectively the activation of the *integer constraint* loss, and the activation of the clamping. This optimization is performed with gradient descent directly on GPUs, with a small additional cost to the inference procedure.

## D  NP-hardness of Coverage Maximization in N-way Alignment

Given the set of possible N-way alignments, the problem of finding the one that maximizes the target coverage is NP-hard. To prove it, we can reduce the NP-hard *set cover* problem (Garey and Johnson, 1979) to the *N-way alignment coverage maximization* problem.

- *Cover set* decision problem (A):

  Let $X = \{x_1, \cdots, x_N\}$ and $C_0 \subset 2^X$. Is there $c^* = (c_1, \cdots, c_K) \in C_0^K$ s.t. $|\cup_k c_k^*| = |X|$?

- *N-way alignment coverage maximization* decision problem (B):

  Let $X = \{x_1, \cdots, x_N\}$ and $C = (C_1, \cdots, C_K) \subset (2^X)^K$. For $p \in \mathbb{N}$, is there $c \in \prod_{k=1}^K C_k$ s.t. $|\cup_k c_k| \geq p$?

A solution of (B) can be certified in polynomial time: we simply compute the cardinal of a union. Any instance of (A) can be transformed in polynomial time and space into a special instance of (B) where all $C_k = C_0$ and $p = |X|$.

## E   Results for fr-en

Table **??** reports the BLEU scores for the reverse direction (fr→en), using exactly the same configuration as in Table 2. Note that since we used the same data split (retrieving examples based on the similarity in English), and since the retrieval procedure is asymmetrical, 4,749 test samples happen to have no match. That would correspond to an extra column labeled "0", which is not represented here.

| Model \N | 1 | 2 | 3 | all |
|---|---|---|---|---|
| size | 2,753 | 1,675 | 12,823 | 17,251 |
| TM[1]-LevT | 57.2 | 57.6 | 61.5 | 60.2 |
| TM[3]-LevT +pt +ra | 58.2 | 59.4 | 64.0 | 62.4 |

Table 7: BLEU scores on the full test set. TM[3]-LevT is improved with pre-training and realignment. All BLEU differences are significant ($p = 0.05$). $p$-values from SacreBLEU paired bootstrap resampling ($n = 1000$).

The reverse direction follows a similar pattern, providing further evidence of the method's effectiveness.

## F   Complementary Analyses

**Diversity and difficulty**   Results in Table 4 show that some datasets do not seem to benefit from multiple examples. This is notably the case for Europarl, News-Commentary, TED2013, and Ubuntu. We claim that this was due to the lack of retrieved examples at training (as stated in §B), of diversity, and the noise in fuzzy matches. To further investigate this issue, we report two scores in Table 8. The first is the increase of bag-of-word coverage

of the target gained by using $N$=3 instead of $N$=1; the second is the increase of noise in the examples, computed as the proportion of tokens in the examples that do not occur in the target. We observe that, in fact, low diversity is often associated with poor scores for TM[3]-LevT, and higher diversity with better performance.

| | cover | | noise | |
|---|---|---|---|---|
| | *test-0.4* | *test-0.6* | *test-0.4* | *test-0.6* |
| ECB | +8.2 | +8.9 | +3.7 | +4.4 |
| EME | +8.9 | +8.5 | +4.0 | +5.0 |
| Epp | +10.5 | +13.7 | +2.2 | +4.0 |
| GNO | **+7.1** | **+6.2** | **+7.6** | **+9.3** |
| JRC | **+7.2** | **+6.8** | +4.8 | +5.2 |
| KDE | +8.0 | **+7.5** | **+6.9** | **+7.8** |
| News | **+7.2** | +12.5 | +2.3 | +5.0 |
| PHP | **+7.2** | **+7.6** | +4.2 | +5.4 |
| TED | +9.4 | +11.4 | +2.9 | +4.6 |
| Ubu | **+5.5** | **+6.0** | **+6.7** | **+8.6** |
| Wiki | +8.0 | +8.0 | +2.5 | +3.6 |
| all | +8.1 | +8.9 | +4.4 | +5.7 |

Table 8: Coverage and noise scores increase. "Difficulty" is highlighted in bold ($< 8.0$ for cover; $> 6.0$ for noise).

**Long run**   All results in the main text were obtained with models trained for 60k iterations, which was enough to compare the various models while saving computation resources. For completeness, we also performed one longer training for 300k iterations for TM[3]-LevT (see Table 9), which resulted in an improvement of around +2 BLEU for each test set. This is without realignment nor pretraining.

| model | | *test-0.4* | *test-0.6* |
|---|---|---|---|
| TM[3]-LevT | | 46.5 | 60.0 |
| | + realign | 46.7 | 60.2 |
| TM[3]-LevT | long | 48.7 | 61.9 |
| | + realign | **48.9** | **62.0** |

Table 9: BLEU score of TM[3]-LevT: 60k iterations; and TM[3]-LevT long: 300k iterations.

**The Benefits of realignment**   Table 10 shows that realignment also decreases the average number of refinement steps to converge. These results suggest that the edition is made easier with realignment.

In Table 11, we present detailed results of the unigram modified precision of LevT, TM[3]-LevT and

| model | | test-0.4 | test-0.6 |
|---|---|---|---|
| TM$^3$-LevT | | 3.55 | 2.07 |
| | +realign | **3.37** | **1.93** |

Table 10: Average number of extra refinement rounds.

TM$^3$-LevT+realign. Using more examples indeed increases copy (+4.4), even though it diminishes copy precision (-1.7). Again we observe the positive effect of realignment, which amplifies the tendency of our model to copy input tokens.

| model | | precision | % units |
|---|---|---|---|
| TM-LevT | *copy* | **87.5** | 64.9 |
| | *gen* | 52.6 | 35.1 |
| TM$^3$-LevT | *copy* | 85.4 | 68.8 |
| | *gen* | 54.9 | 31.2 |
| +realign | *copy* | 85.8 | **69.3** |
| | *gen* | 54.7 | 30.7 |

Table 11: Modified precision of copy vs. generated unigrams of LevT vs. TM$^3$-LevT.

**COMET scores**  We compute COMET scores (Rei et al., 2020) separately for each domain with default wmt20-comet-da similarly to Table 4 (see Table 12). We observe that the basic version of TM$^3$-LevTunderperforms TM$^1$-LevT; we also see a great variation in the scores. A possible explanation can be a fluency decline when using multiple examples, which is not represented by the precision scores computed by BLEU. The improved version, using realignment and pre-training, confirms that adding more matches is overall beneficial for MT quality.

**Per-domain ablation study**  Table 13 details the results of our ablation study separately for each domain.

**Illustration**  A full inference run is in Table 14, illustrating the benefits of considering multiple examples and realignment. Even though the realignment does not change here the final output, it reduces the number of atomic edits needed to generate it, making the inference more robust.

| | | ECB | EME | Epp | GNO | JRC | KDE | News | PHP | TED | Ubu | Wiki | all |
|---|---|---|---|---|---|---|---|---|---|---|---|---|---|
| *test-0.4* | TM$^1$-LevT | 33.0 | 43.1 | 39.9 | 56.3 | 70.7 | 37.4 | -2.0 | -39.6 | -0.8 | 41.6 | -9.9 | 24.5 |
| | TM$^2$-LevT | 27.2 | 42.0 | 31.1 | 48.0 | 64.4 | 32.6 | -10.8 | -42.7 | -8.7 | 35.3 | -15.7 | 18.4 |
| | TM$^3$-LevT | 27.3 | 42.1 | 26.8 | 51.5 | 64.2 | 33.5 | -10.1 | -39.9 | -14.8 | 38.6 | -16.3 | 18.4 |
| | +pre-train | 30.6 | **44.7** | 38.7 | **57.9** | 67.8 | **37.9** | -6.2 | **-35.3** | 2.7 | 41.5 | **-5.0** | **25.0** |
| | +realign | 31.0 | **45.1** | 32.0 | 53.2 | 66.4 | 34.9 | -10.9 | -39.0 | -11.9 | 41.1 | -10.8 | 21.0 |
| | +both | **33.7** | **46.4** | **42.4** | **59.9** | 69.9 | **40.1** | **-1.0** | **-33.0** | 5.1 | **43.5** | **-3.7** | **27.5** |
| *test-0.6* | TM$^1$-LevT | 51.7 | 53.9 | 56.9 | 65.3 | 85.4 | 37.0 | -3.0 | -17.0 | 48.5 | 57.6 | 64.5 | 45.5 |
| | TM$^2$-LevT | 51.2 | **54.2** | 56.3 | 64.2 | 82.5 | 34.6 | -9.0 | **-15.9** | 46.1 | 55.5 | 61.6 | 43.7 |
| | TM$^3$-LevT | 50.9 | **55.8** | 54.1 | 65.1 | 81.1 | 33.6 | -9.8 | **-16.2** | 41.1 | 57.4 | 61.7 | 43.1 |
| | +pre-train | **54.6** | **56.5** | **58.0** | **68.2** | 84.7 | **41.5** | -4.3 | **-11.8** | **48.9** | **62.7** | **65.6** | **47.7** |
| | +realign | **53.0** | **56.4** | 56.3 | 65.4 | 83.6 | 34.3 | -7.2 | **-16.1** | 42.8 | **58.3** | 63.7 | 44.6 |
| | +both | **55.7** | **57.4** | **58.8** | **70.5** | **85.8** | **43.2** | -4.4 | **-10.6** | **49.4** | **62.3** | **67.7** | **48.7** |

Table 12: Per domain COMET scores (x 100) for TM$^n$-LevT and variants. Bold for scores better than TM$^1$-LevT.

| | ECB | EME | Epp | GNO | JRC | KDE | News | PHP | TED | Ubu | Wiki | all |
|---|---|---|---|---|---|---|---|---|---|---|---|---|
| *test-0.4* | | | | | | | | | | | | |
| TM$^3$-LevT | 53.9 | **55.6** | **34.2** | 60.7 | **66.0** | **53.5** | **20.4** | 33.0 | **28.6** | 47.5 | **32.8** | **46.5** |
| -sel | **54.5** | **55.6** | 32.7 | **61.2** | 65.9 | 52.2 | 19.5 | **33.3** | 27.7 | **48.1** | 31.3 | 46.2 |
| -delx | 52.2 | 53.4 , | 31.8 | 58.7 | 64.0 | 52.0 | 19.6 | 31.2 | 27.5 | 46.8 | 31.5 | 44.8 |
| -rd-del | 49.7 | 47.6 | 22.2 | 48.2 | 56.7 | 38.8 | 13.2 | 29.5 | 16.9 | 32.2 | 21.4 | 38.6 |
| -mask | 53.4 | 54.7 | 33.7 | 58.9 | 65.3 | 52.4 | 20.3 | 33.1 | 27.5 | 47.1 | 32.2 | 46.0 |
| -dum-plh | 50.8 | 43.3 | 32.7 | 45.6 | 61.2 | 42.4 | 21.0 | 30.9 | 24.3 | 35.6 | 28.2 | 41.0 |
| -indep-align | 51.5 | 52.4 | 29.7 | 53.8 | 60.7 | 47.2 | 16.9 | 30.4 | 21.8 | 39.0 | 28.4 | 42.6 |
| *test-0.6* | | | | | | | | | | | | |
| TM$^3$-LevT | **64.2** | **68.0** | 49.4 | **73.0** | **76.4** | **60.1** | 21.2 | **39.6** | 52.2 | **60.1** | 61.6 | **60.1** |
| -sel | 63.8 | 67.5 | **50.0** | 71.2 | 76.3 | 60.0 | **21.5** | 39.1 | **54.2** | 59.7 | **62.2** | 60.0 |
| -delx | 62.1 | 66.7 | 47.7 | 70.6 | 75.0 | 58.8 | 20.1 | 37.9 | 53.5 | 58.4 | 60.9 | 58.6 |
| -rd-del | 58.4 | 59.6 | 39.7 | 59.0 | 67.6 | 47.7 | 16.9 | 35.0 | 39.8 | 44.1 | 47.4 | 51.9 |
| -mask | 63.1 | 65.3 | 49.1 | 69.4 | 74.2 | 58.2 | 21.8 | 38.6 | 50.9 | 58.7 | 59.7 | 59.0 |
| -dum-plh | 57.3 | 55.5 | 44.9 | 51.8 | 68.5 | 44.5 | 20.2 | 35.7 | 42.6 | 38.7 | 50.7 | 50.9 |
| -indep-align | 60.6 | 64.0 | 46.6 | 64.3 | 71.9 | 55.5 | 18.6 | 35.6 | 44.6 | 51.6 | 56.0 | 56.4 |

Table 13: Ablation study. We report BLEU scores for various settings. -sel: no random selection noise ($\gamma$=0); -delx: no extra deletion loss; -rd-del: no random deletion ($\beta$=0); -mask: no random mask ($\delta$=0); -dum-plh: null probability to start with $\mathbf{y}_{post\cdot del}=\mathbf{y}_*$ ($\alpha = 0$); -indep-align: the alignments are performed independently.

| | |
|---|---|
| **src**: | The swf _ setfont ( ) sets the current font to the value given by the fontid parameter . |
| **tgt**: | swf _ setfont ( ) remplace la police courante par la police repérée par l ' identifiant fontid . |

| | *LevT* |
|---|---|
| $\mathbf{y}^{del}$: | sw● f _ ~~fon● t●~~ ~~size~~ ( ) remplace la ~~taille~~ de ~~la~~ police par la ~~taille~~ ~~size~~ . |
| $\mathbf{y}^{plh}$: | sw● f _+2 ( ) remplace la+1 de police+1 par la+5 . |
| $\mathbf{y}^{tok}$: | sw● f _ set● police ( ) remplace la police de police courante par la valeur de paramètre ti● d . |
| $\mathbf{y}^{del}$: | sw● f _ set● police ( ) remplace la police de police courante par la valeur de paramètre ti● d . |
| $\mathbf{y}^{plh}$: | +1 sw● f _ set● police ( ) remplace la police de police courante+1 par la valeur de paramètre+1 ti● d . |
| $\mathbf{y}^{tok}$: | Le sw● f _ set● police ( ) remplace la police de police courante donnée par la valeur de paramètre fon● ti● d . |
| **hyp**: | Le swf _ setpolice ( ) remplace la police de police courante donnée par la valeur de paramètre fontid . |

| | TM$^2$-LevT |
|---|---|
| $\mathbf{y}^{del}$: | sw● f _ ~~fon● t●~~ ~~size~~ ( ) remplace la ~~taille~~ ~~de~~ ~~la~~ police par ~~la~~ ~~taille~~ ~~size~~ . |
| | sw● f _ ~~defin●~~ ~~e●~~ font ( ) définit la police ~~fon●~~ ~~t●~~ ~~name~~ ~~et~~ ~~lui~~ ~~affecte~~ l ~~'~~ ~~identi●~~ ~~fiant~~ fon● ti● d . |
| $\mathbf{y}^{plh}$: | sw● f _+2 ( ) remplace la police+6 par+4 . |
| | sw● f _+1 font ( ) définit la police+9 fon● ti● d . |
| $\mathbf{y}^{cmb}$: | sw● f _ ( ) remplace la police par : |
| | sw● f _ font ( ) définit la police fon● ti● d . |
| $\mathbf{y}^{tok}$: | sw● f _ set● font ( ) remplace la police actuelle à à la valeur donnée par le paramètre fon● ti● d . |
| $\mathbf{y}^{del}$: | sw● f _ set● font ( ) remplace la police actuelle ~~à~~ à la valeur donnée par le paramètre fon● ti● d . |
| $\mathbf{y}^{plh}$: | sw● f _ set● font ( ) remplace la police actuelle+1 la valeur donnée par le paramètre fon● ti● d . |
| $\mathbf{y}^{tok}$: | sw● f _ set● font ( ) remplace la police actuelle à la valeur donnée par le paramètre fon● ti● d . |
| **hyp**: | swf _ setfont ( ) remplace la police actuelle à la valeur donnée par le paramètre fontid . |

| | TM$^3$-LevT |
|---|---|
| $\mathbf{y}^{del}$: | sw● f _ ~~fon● t●~~ ~~size~~ ( ) remp● place la ~~taille~~ ~~de~~ ~~la~~ police par ~~la~~ ~~taille~~ ~~size~~ . |
| | sw● f _ ~~defin●~~ ~~e●~~ font ( ) définit la police ~~fon●~~ ~~t●~~ ~~name~~ ~~et~~ ~~lui~~ ~~affecte~~ l ~~'~~ ~~identi●~~ ~~fiant~~ fon● ti● d . |
| | sw● ~~f●~~ ~~text~~ - set● font ( ) remplace la police courante par ~~font~~ . |
| $\mathbf{y}^{plh}$: | sw● f _+2 ( ) remplace la police+4 par+5 . |
| | sw● f _+1 font ( ) définit la police+7 ti● d . |
| | sw●+2 set● font ( ) remplace la police courante+4 par+5 . |
| $\mathbf{y}^{cmb}$: | sw● f _ ( ) remplace la police par : |
| | sw● f _ font ( ) définit la police ti● d . |
| | sw● set● font ( ) remplace la police courante ~~par~~ . |
| $\mathbf{y}^{tok}$: | sw● f _ set● font ( ) remplace la police courante au valeur donnée par le fon● ti● d . . . |
| $\mathbf{y}^{del}$: | sw● f _ set● font ( ) remplace la police courante ~~au~~ valeur ~~donnée~~ par le fon● ti● d ~~. . .~~ |
| $\mathbf{y}^{plh}$: | +1 sw● f _ set● font ( ) remplace la police courante+2 valeur+1 par le+1 fon● ti● d . |
| $\mathbf{y}^{tok}$: | Le sw● f _ set● font ( ) remplace la police courante à la valeur donnée par le paramètre fon● ti● d . |
| **hyp**: | Le swf _ setfont ( ) remplace la police courante à la valeur donnée par le paramètre fontid . |

| | TM$^3$-LevT + *realign* |
|---|---|
| $\mathbf{y}^{del}$: | sw● f _ ~~fon● t●~~ ~~size~~ ( ) remp● place la ~~taille~~ ~~de~~ ~~la~~ police par ~~la~~ ~~taille~~ ~~size~~ . |
| | sw● f _ ~~defin●~~ ~~e●~~ font ( ) définit la police ~~fon●~~ ~~t●~~ ~~name~~ ~~et~~ ~~lui~~ ~~affecte~~ l ~~'~~ ~~identi●~~ ~~fiant~~ fon● ti● d . |
| | sw● ~~f●~~ ~~text~~ - set● font ( ) remplace la police courante par ~~font~~ . |
| $\mathbf{y}^{plh}$: | sw● f _+2 ( ) remplace la police+4 par+5 . |
| | sw● f _+1 font ( ) définit la police+8 ti● d . |
| | sw●+2 set● font ( ) remplace la police courante+3 par+5 . |
| $\mathbf{y}^{cmb}$: | sw● f _ ( ) remplace la police par . |
| | sw● f _ font ( ) définit la police ti● d . |
| | sw● set● font ( ) remplace la police courante par . |
| $\mathbf{y}^{tok}$: | sw● f _ set● font ( ) remplace la police courante au valeur donnée par le paramètre fon● ti● d . |
| $\mathbf{y}^{del}$: | sw● f _ set● font ( ) remplace la police courante ~~au~~ valeur donnée par le paramètre fon● ti● d . |
| $\mathbf{y}^{plh}$: | +1 sw● f _ set● font ( ) remplace la police courante+2 valeur donnée par le paramètre fon● ti● d . |
| $\mathbf{y}^{tok}$: | Le sw● f _ set● font ( ) remplace la police courante à la valeur donnée par le paramètre fon● ti● d . |
| **hyp**: | Le swf _ setfont ( ) remplace la police courante à la valeur donnée par le paramètre fontid . |

Table 14: Examples of full inference of several models on a test sample from *test-0.4-PHP* (sample n°571). Copied parts are in red.