# OpenReview forum: "Towards Example-Based NMT with Multi-Levenshtein Transformers"
_EMNLP/2023/Conference — EMNLP 2023 Main_

### Official Review · Reviewer_Fysf · 2023-08-04

**Soundness:** 5

**Excitement:**

4: Strong: This paper deepens the understanding of some phenomenon or lowers the barriers to an existing research direction.

**Missing References:**

- Related to Section 4.2, a long time ago https://aclanthology.org/2005.eamt-1.23/ actually used MSA to bootstrap word alignments. A mention to it when referencing "This problem can be solved using Dynamic Programming (DP) techniques similar to Multiple Sequence Alignment (MSA)" would be good.


**Paper Topic And Main Contributions:**

Summary: This paper adapts a retrieval based levenshtein transformers (LevT) to utilize translation memory in machine translation (TM3-LevT). The main contributions for this paper:

- introducing the combination layer in LevT (layers here also refers to "edit operations")
- using maximal cover policy instead of levenshtein distance (Section 4); proposing a simplified method to align TM matches to target sentence and using the alignments to determine edit policy


Strength:
- Interesting simplified approach in Section 4.2 for alignment capitalizing on the fact that the no. of n-way TM matches are small. An old-timer MT practitioner from phrase-based MT era might simply added IBM model 2 or model 4 alignments to the translation inputs, e.g. https://aclanthology.org/W16-2206
- [major] following up on the experiments results, highlighting limitations and proposing methods to overcome the limitations
- [major] Rigorous work done to show comparison of proposed architecture against (i) vanillia auto-regressive model, (ii) default LevT model, (iii) various ablation of components in the proposed architecture and (iv) models with refinements made after initial experiments


Weakness:
- [minor] The simplified alignment method, word / token alignment is extremely hard in MT. While the simplified approach + realignment efforts (Appendix C) is commendable, it'll be make the paper even better if the proposed alignment is compared to well-studied and battle-proofed IBM models, e.g. from mgiza++. (Putting this down as a minor weakness because the older software/tools that runs IBM models are not that well-documented but might not be that accessible to researchers who are unfamiliar with the tools)
- [minor] WMT14 only evaluation is generally frowned upon in recent MT works but in the case of lexically constraint MT or TM-MT, that seems to be the only WMT14 evaluation set that are comparable to previous work. Also, usually people also compare results to the EN-DE portion of WMT14 that comes with a dictionary https://aclanthology.org/P19-1294/ but this wasn't present in the paper (I understand the choice of dataset might be arbitrary so putting this down as a minor weakness).

**Questions For The Authors:**

- Is there an attempt to increase the TM matches so that (2) assumption in Section 4.2 becomes intractable? "For each training sample (x, y), we retrieve up to 3 in-domain matches", does that mean that N=3 and you've set k=10 according to the settings in the Appendix A. Is my understanding of N and k correct?

- Would the source for the proposed method be open sourced?

- Will the splits of the train/dev/test sets be release?




**Reasons To Accept:**

Well-written and clear presentation and writing. Thorough work done to the proposed architecture. The strengths of the paper definitely outweighs the weakness, see above for details.



**Reasons To Reject:**

Mainly the experiment evaluation being carried out in one direction of translation and for a single language from an older test sets but these are minor weakness given precedence that this particular topic has always been evaluated this way. More details, see weaknesses above


**Reproducibility:**

4: Could mostly reproduce the results, but there may be some variation because of sample variance or minor variations in their interpretation of the protocol or method.

**Reviewer Confidence:**

4: Quite sure. I tried to check the important points carefully. It's unlikely, though conceivable, that I missed something that should affect my ratings.

**Typos Grammar Style And Presentation Improvements:**

**Stylistic:**

- Section 3, line 181 "The latter component ...", if you are referring to S (sequence embedding), avoid ambiguity in the writing, "The latter component" -> "The sequence embedding ..."
- "Conclusion and outlook " -> "Conclusion and Outlook"
- Also keep the headings of \section{} and \subsection{} consistent, sometimes it's titlecase, sometimes is capitalized only on the first word, choose one. Personal preference is title case for all \section{} and caps for only first word for \subsection{}

**Content (Please add to revision)**:

- Please put in footnote or experimental setup in appendix the version of sacrebleu and also the version and model used in reported comet score
- Please also report chrf scores from sacrebleu for Table 2, you can put it in the appendix if there's space constraints


**Content Suggestions:**

- Take a look at Section 2.2 of https://aclanthology.org/W16-4618 to get the mgiza software working in a quick/fast manner, use those alignments to get  (V, V∗, E) and see if it works better than the proposed method. Alternatively, there's https://www.nltk.org/api/nltk.align.html but it's rather slow.

- The "excitement" score would be a 5 if this weakness is resolve by re-running the Table 8 experiments on:
   - (reverse direction) FR-EN on WMT14
   - (comparable to previous LevT work) EN-DE with lexicon on WMT14
   - Try a lower resource language, some previous LevT papers put additional language experiments in the appendix to be supporting evidence that methods work beyond a single language pair while keeping the narrative in the main text simple.

---

> ### Author Rebuttal · Authors · 2023-08-29
>
> We thank you for your review.
>
> Note that both code and data split indices will be released on github upon publication with detailed instructions (cf. *lines 89-90*).
>
> **Some remarks**:
>
> mgiza++, nltk.align are for bilingual alignments. In our case, we perform an exact monolingual lexical N-way alignment in the target language only (the target-side TMs aligned with the reference). Only identical words can be aligned -- which means that the lexical alignment model is trivial and does not need to be learned.
>
> Thanks for the additional reference to MSA for word alignment.
>
> It is true that we only experimented with one pair (en-fr), while we could have used WMT14 and (en-de) to be in line with recent MT works.
> We explain this choice with two reasons:
> * It makes it comparable to some other works using the same multidomain data, like Xu et al., 2023, who used a similar setup.
> * We can already widely explain the observed differences in performances across the multiple domains with our data (see Appendix E).
> We intend to also reproduce experiments for the other language direction, and eventually other language pairs.
>
>
> **Answers to questions raised**:
>
> * Your understanding of N and k is correct. As for N, we did not perform experiments with higher value since the computational cost of the alignment is exponential with respect to N. The training would become too slow, while still theoretically feasible. A future work could consist of further improving the complexity of the N-way alignment.
> * The versions of SacreBleu and Comet used in the experiments are documented in Appendix A.
> * We will add chrf scores from SacreBleu in the appendix as suggested

---

### Official Review · Reviewer_k9eN · 2023-08-05

**Soundness:** 3

**Excitement:**

3: Ambivalent: It has merits (e.g., it reports state-of-the-art results, the idea is nice), but there are key weaknesses (e.g., it describes incremental work), and it can significantly benefit from another round of revision. However, I won't object to accepting it if my co-reviewers champion it.

**Missing References:**

See the question for the reference that could be useful to your work.

**Paper Topic And Main Contributions:**

The Levenshtein transformer (TM-LevT) of Gu et al. (2019) is an encoder-decoder model which, given a source sentence, predicts edits including insertion and deletion that are applied to an initial translation and generate a revision of the output. The initial translation corresponds to a match from a TM (or an empty output). The paper proposes simultaneously extends TM-LevT to TM^N -LevT, which the goal is to handle several initial translations retrieved from TMs. This sounds pretty straightforward but it is actually not obvious at all, involving multiple algorithmic and computational challenges. The authors present preliminary experiments, showing that using TM^N -LevT
gives better translation accuracy to TM -LevT. I could not follow all the details because the paper itself is extremely technical to me. But I like the novelty of the work a lot. I think it is the strongest point of the work. Meanwhile, there is clear concern with the complexity of the framework, and whether the baseline is strong enough for a good conclusion of how useful the framework is.

**Questions For The Authors:**

For the learning part, I think the idea of shuffling top_k fuzzy matches during training (i.e. shuffle 3 examples from top-10 instead of using top-3 references) maybe useful for training your model.
Reference:
Improving Robustness of Retrieval Augmented Translation via Shuffling of Suggestions, Cuong Hoang et al., 2023 https://arxiv.org/abs/2210.05059

[Updated after author response] - I thank authors for their response.

**Reasons To Accept:**

* Novelty of the idea of having a combination of N fuzzy matches for generating the final translation from a Levenshtein Transformer is the strongest point of the paper in my opinion. Using N fuzzy matches is an obvious approach to improve translation accuracy of Retrieval-Augmented Machine Translation in general. Nonetheless it is not obvious at all how it works (especially both training and decoding part) with Levenshtein Transformer. The paper shows that there is a way to do so, and there is some benefit (i.e. improving in translation accuracy)

**Reasons To Reject:**

While I like the work a lot, I think there are two main critical feedbacks of the paper:
* The complexity of the framework is not straightforward at all to implement, let alone it is very expensive to train the model (btw how is about the complexity of the inference?). I tried to pay very close attention to the details but I give up at some point because it is not obvious at all some details of the paper. While I recommend the authors to revise the writing a bit to make sure the technical part of the paper is clear, releasing the implementation if possible is very useful for this work.
* The baseline for the experiment is not that strong, for two reasons. First, the baseline does not enrich source sentence with fuzzy matches, which would make a much stronger baseline (I even suspect the improvement of the proposed method will be very little in that case). Second, the baseline is with NAT, which is not strong compared to using auto-regressive model. So it is hard to convince the usefulness of the framework.

**Reproducibility:**

2: Would be hard pressed to reproduce the results. The contribution depends on data that are simply not available outside the author's institution or consortium; not enough details are provided.

**Reviewer Confidence:**

3: Pretty sure, but there's a chance I missed something. Although I have a good feel for this area in general, I did not carefully check the paper's details, e.g., the math, experimental design, or novelty.

**Typos Grammar Style And Presentation Improvements:**

Unfortunately the framework is very technical. I don't blame the paper at all about this, just I think some of the following things need to be clear:
* Extending TM-LevTto -> add extra space (line 72)
* which extends TM-LevTto -> add extra space (line 68)
* an increase in translation performance -> performance can mean different things. I guess what you meant here is the accuracy.
* As editing operations are not observed in 132 the training data, LevT resorts to Imitation Learn- 133 ing, based on the generation of decoding configu- 134 rations for which the optimal prediction is easy to 135 compute -> This detail is important but the paper is not self-contained at this point.
* Line 177 - Denoting y = (y1, · · · , yN ), N retrieved instances, yn,i is encoded as Eyn,i + Pi + Sn -> I supposed E is the embedding of the token, P is the positional encoding of the token, Sn is the encoding of the source sentence.
Sn is explained in line 183 but it is very vague what it is. Plus why would we sum all of them (Eyn,i + Pi + Sn as they have different shapes) and why would the lenth of S is N + 1 instead of N?
* In Figure 2, please explain in the description about del (delete), plh (insert), cmb (combination) and tok (prediction).
* Line 195: Why A has dimension of 2xd_model instead of d_model?
* Line 195 and line 198: each deleted token y_{n, i}^{del} and each insertation y_{n, i}^{plh} -> I don't think I could follow the difference between these two tokens. They should be the same???

I could not follow in detail section 3.3 and 4. I think these are very technical and I hope the authors will make them more readable.

---

> ### Author Rebuttal · Authors · 2023-08-29
>
> We thank you for the review. We will fix factual errors and typos. We also reckon that for the sake of space, several technical details had to be compressed. Having one extra page will give us the opportunity to make our presentation less dense in the revised version.
>
> Thank you also for suggesting the idea of shuffling top_k matches. We will consider it in our future work, along with other approaches to improve retrieval (e.g. with contrastive examples, etc).
>
> We noted worries about the reproducibility. Note that both code and data split indices will be released on github upon publication with detailed instructions (cf. *lines 089-090*).
>
> Section 3.3 and 4 have been reported to be hard to grasp. Admittedly, they are very technical and hard to clearly synthesize. We will do our best to make them more readable. We were also suggested to add an illustration of the alignment process.
>
> **Some remarks**:
>
> We indeed did not enrich the source side with the TM examples due to practical reasons. In fact, it inevitably yields longer inputs and smaller batch sizes. Large batch sizes are essential in NAT setups, so this would have to be compensated with more GPUs or cumulative gradients, which would likely have increased the training time in our experiments. Note that in principle, mLevT could also benefit from an enriched source side.
>
> Nonetheless, we agree that there is still a gap between AR and NAT models (cf. results in Table 4). Our work aims to improve TM-augmented NAT and close this gap (which it partially does), also for the sake of explaining (more precisely tracing back) some translation outputs, which an AR system cannot do.
>
> **Answers to questions raised**:
>
> * Inference complexity: Our method yields a higher computational cost at training due to the alignment step, which has an algorithmic complexity of $O(k^N)$. In practice, it approximately doubles the training time. As for inference, the first cycle of edition only has one extra combination step compared to LevT. So it barely increases the latency in theory and in practice. This will be clarified in the revised version.
> * *line 177*: $E_{y_{n,i}}$, $P_i$ and $S_n$ all have dimension $d_{model}$ since the written indices select the columns in matrices $E$, $P$ and $S$. Moreover, $S$ has a role of discrimination between the multiple TMs. $S_n$ thus identifies TM $y_n$. As for the extra dimension, $S_N$ identifies the combined sentence obtained through combination ($y^{tok}$), which undergoes refinement. We can add a sentence explaining this point that is not obvious.
> * *line 195*: Matrix $A$ has a $2 \times d$ shape since the decision of deletion is formulated as a classification problem with 2 classes, and not a logistic regression with a decision threshold. This is the choice made in the original LevT paper.
> * *line 195-198*: $y^{del}$ is the TM sentences before deletion while $y^{plh}$ is the result of applying deletions, and before insertion of placeholders.

---

### Official Review · Reviewer_4qBu · 2023-08-07

**Soundness:** 3

**Excitement:**

3: Ambivalent: It has merits (e.g., it reports state-of-the-art results, the idea is nice), but there are key weaknesses (e.g., it describes incremental work), and it can significantly benefit from another round of revision. However, I won't object to accepting it if my co-reviewers champion it.

**Paper Topic And Main Contributions:**

The paper extends the Levenshtein Transformer of Gu et al. 2019 to work on editing multiple examples from the translation memory. To this end, the set of edit operations is expanded with combination, which selects tokens from each example sentence that will be kept in the combined sentence. To train the model, the training via imitation learning is modified so that the expert policy tracks a heuristically optimal N-way alignment between translation memory examples and the target. The paper reports improvements from this model over Levenshtein Transformer from a single example.

**Questions For The Authors:**

* Lines 32-33: "the autoregressive nature of the decoder can make the process..." What does it have to do with the autoregressive decoder? One can have an edit-based MT system with an autoregressive decoder.

**Reasons To Accept:**

* The paper presents an interesting extension of Levenshtein Transformer to a more realistic scenario where multiple examples from the translation memory are found relevant to the translation at hand.
* The solution (the combination operation and the modification to the training via imitation learning) is novel and technically interesting.
* The solution leads to performance improvements over Levenshtein Transformer.

**Reasons To Reject:**

I find the paper not very well structured and not clearly written.

  + The model of Gu et al. 2019, to which this work makes modifications, should be explained in all necessary detail. Without this context, the paper is hard to follow. Also, presenting Gu et al. 2019 first would make it clear which modifications and additions this paper makes.

  + The combination operation is not clear. Ostensibly (Figure 3), combination also performs deletions (just mark a token to not be selected for the combination). So why do we need deletion before combination? Combination is one of the main contributions of the paper, it has to be explained accordingly.

  + Section 3.3 is hard to understand. The problem is not in the subject matter (imitation learning roll-ins), but the content presentation. Figure 4 needs clarification. Also, what exactly are the contributions of this paper in this section?

  + The heuristic N-way alignment (Section 4.2) needs clarification and examples. It would also be helpful to see examples of when the heuristic alignment fails to recover the optimal alignment.

  + I don't understand how expert roll-outs are computed in non-initial states. A detailed explanation of how the training works is in order.

  + Generally, here and there, there is a lot of notation that is not properly introduced.

**Reproducibility:**

3: Could reproduce the results with some difficulty. The settings of parameters are underspecified or subjectively determined; the training/evaluation data are not widely available.

**Reviewer Confidence:**

3: Pretty sure, but there's a chance I missed something. Although I have a good feel for this area in general, I did not carefully check the paper's details, e.g., the math, experimental design, or novelty.

---

> ### Author Rebuttal · Authors · 2023-08-29
>
> We thank you for the review. We will fix factual errors and typos. We also reckon that for the sake of space, several technical details had to be compressed. Having one extra page will give us the opportunity to make our presentation less dense in the revised version.
>
> We noted worries about the reproducibility. Note that both code and data split will be released on github upon publication with detailed instructions (this is mentioned in *lines 089-090*).
>
> Section 3.3 and 4 have been reported to be hard to grasp. We will do our best to make them more readable, and to better illustrate the alignment process. We will also change subsection "Learning" to "Imitation Learning". Please, do not hesitate to share ideas to improve these sections.
>
> **Some remarks**:
>
> We give a high-level presentation of Gu et al.'s LevT in section 2.2; in the revised version, we will better expand it and also illustrate the commonalities and differences with our own approach in Figure 2
>
> **Answers to questions raised**:
> * *Lines 32-33*: While point (a) is linked to the autoregressive nature, (b) is due to the generative nature (i.e. to the fact translation is generated from scratch). You are right to point out that an edit-based model could also be auto-regressive -- we will rephrase this sentence.
>
> * About the *combination*:
> The combination operation can yield to deletions when it selects a placeholder at some position. In the ideal case (when the deletion model is error-free), this would not happen. But since there might be mistakes with undeleted tokens in the deletion component, the combination can act as a safety layer of deletion. But it is more an implication in the design choice than a "second deletion". An initial deletion/insertion is still necessary to generate aligned sequences, from which the combination module selects the right sequence of positions (see Figure 3).
> We will add an explanation to clarify this point.
>
> * About *training* (section 3.3), except for the initial state and for state (5), we claim the computation of optimal edition is "obvious" (see *line 288*). This is because training examples are generated by randomly perturbing sequences in a way that we can easily reverse. For instance, when deleting random tokens with $\pi^{rnd.del.N}$, the optimal insertion is to put back placeholders where the deletions occurred.
> We will clarify this point.

---

### Meta-Review · Area_Chair_7khh · 2023-09-19

**Recommendation:** 4

**Metareview:**

The reviewers agree that the paper presents a novel and interesting contribution and agree on a good soundness score. There were a couple of issues raised by the reviewers, that were also acknowledged by the authors, and for some I want to go a bit into more detail here.

Some reviewers complained about the paper being hard to follow and too technical at some points. While I understand the position of the reviewer, the exact mathematical notation is necessary for such complex systems and it actually makes the paper stronger, although perhaps more difficult to follow for a wider audience. Nevertheless I would encourage the authors to add some more intuitive explanation for the most complex concepts. Maybe including a fully detailed example in the appendix might help?

Another reviewer also complained about older test sets being used in the evaluation. Regretfully this is an extended "evil" among our community and the authors defended their decision arguing comparison with previous work. In the rebuttal period the authors provided evidence of generalization to another language pair. These results should definitely be included in the final version to make the paper stronger.

All in all, the paper presents an interesting contribution. In its current form there are some issues in the presentation that have been pointed out by the reviewers, but all can be addressed in a final version of the paper, making use of the extra page of content or expanding the appendix if necessary.

---

### Decision · Program_Chairs · 2023-10-07

**Decision:**

Accept-Main

**Comment:**

The reviewers agree that the paper presents a novel and interesting contribution and agree on a good soundness score. There were a couple of issues raised by the reviewers, that were also acknowledged by the authors, and for some I want to go a bit into more detail here.

Some reviewers complained about the paper being hard to follow and too technical at some points. While I understand the position of the reviewer, the exact mathematical notation is necessary for such complex systems and it actually makes the paper stronger, although perhaps more difficult to follow for a wider audience. Nevertheless I would encourage the authors to add some more intuitive explanation for the most complex concepts. Maybe including a fully detailed example in the appendix might help?

Another reviewer also complained about older test sets being used in the evaluation. Regretfully this is an extended "evil" among our community and the authors defended their decision arguing comparison with previous work. In the rebuttal period the authors provided evidence of generalization to another language pair. These results should definitely be included in the final version to make the paper stronger.

All in all, the paper presents an interesting contribution. In its current form there are some issues in the presentation that have been pointed out by the reviewers, but all can be addressed in a final version of the paper, making use of the extra page of content or expanding the appendix if necessary.